# Biosynthesis of Novel Ascorbic Acid Esters and Their Encapsulation in Lignin Nanoparticles as Carriers and Stabilizing Systems

**DOI:** 10.3390/ijms24109044

**Published:** 2023-05-20

**Authors:** Eliana Capecchi, Davide Piccinino, Chiara Nascimben, Elisabetta Tomaino, Natalia Ceccotti Vlas, Sofia Gabellone, Raffaele Saladino

**Affiliations:** 1Department of Biological and Ecological Sciences (DEB), University of Tuscia, Via San Camillo de Lellis, 01100 Viterbo, Italy; e.capecchi@unitus.it (E.C.); chiara.nascimben@studenti.unitus.it (C.N.); e.tomaino@unitus.it (E.T.); natalia.ceccottivlas@studenti.unitus.it (N.C.V.); 2Istituto Romagnolo per lo Studio dei Tumori “Dino Amadori”—IRST-IRCCS, Via Piero Maroncelli 40, 47014 Meldola, Italy

**Keywords:** ascorbic acid esters, lipase, heterogenous catalysis, encapsulation, lignin nanoparticles

## Abstract

A dual-target strategy was designed for the application of lignin nanoparticles in the lipase mediated biosynthesis of novel 3-*O*-ethyl-L-ascorbyl-6-ferulate and 3-*O*-ethyl-L-ascorbyl-6-palmitate and in their successive solvent-shift encapsulation in order to improve stability and antioxidant activity against temperature and pH-dependent degradation. The loaded lignin nanoparticles were fully characterized in terms of kinetic release, radical scavenging activity and stability under pH 3 and thermal stress (60 °C), showing improved antioxidant activity and high efficacy in the protection of ascorbic acid esters from degradation.

## 1. Introduction

Ascorbic acid (**AA**, Vitamin C) is a water-soluble vitamin produced by the metabolism of plants and animal species [1]. It is characterized by a glucose-like structure comprising six carbon atoms, two of them which are involved in the ene-diol moiety responsible for the reducing activity [2]. AA is used in cosmeceutical, pharmaceutical and nutraceutical applications at high concentration (up to 15% *w*/*w*) as potent antioxidant additive to contrast skin aging processes and to protect the cell against UV-induced photo-damage, promoting the differentiation of keratinocytes and decreasing the biosynthesis of melanin [3]. In addition, it showed repair tissue’s activity, activation of the immune system, neo synthesis of collagens, and production of specific neurotransmitters [4,5,6,7].

Unfortunately, **AA** is susceptible to oxidative degradation showing low stability under both aerobic and anaerobic conditions [8,9], with concomitant loss of biological activity and emergence of unpleasant Maillard browning side processes [10]. The degradation of **AA** involves the oxidation to dehydroascorbic acid (DHA), followed by nucleophilic ring-opening of the lactone ring to yield di-ketogulonic acid (DKG). Finally, DKG releases CO_2_ and L-xylosone (L-XY), 2-furanoic acid (2-AF), and furfural (FU) (Figure 1). This process is associated with change of the color, smell, and gas generation, as well as loss of homogeneity of formulate.

In order to improve the stability of **AA**, appropriate chemical and physical conditions have been selected for the control of solubility, viscosity, polarity of the medium and pH range, which was occasionally associated with emulsifying agents and additives [11,12].

Recent approaches to improve the stability of **AA** encompass the synthesis of stable derivatives or, in alternative, the protection of the molecule by encapsulation technology. In the first case, **AA** is alkylated with alcohols or, in alternative, esterified with carboxylic acids [13,14]. In the second case, it was encapsulated inside micro- and nano carriers from natural (i.e., chitosan, alginate, phospholipids) [15] and synthetic (i.e., acrylic compound [16]) origins [17].

Renewable polyphenols, such as lignin, may be applied for the encapsulation and controlled release of low stable natural substances [18]. Lignin is produced in miles scale of tons annually as a byproduct from the biorefinery and pulp industries [19]. The commercial interest for this polymer has grown in the last year in the context of green chemistry and circular economy [20]. Procedures for the large-scale preparation of lignin nanoparticles (LNPs) are reported in the literature [21]. Compared to amorphous lignin, after the nano-structuration process, LNPs showed improved antioxidant [22], UV shielding [23], and anti-microbial properties, as a consequence of the π–π interactions between the aromatic subunits of the polymer [24]. Examples of encapsulation of bioactive natural substances inside LNPs are reported, including the encapsulation of resveratrol [25] and curcumin [26]. LNPs have been also applied in the stabilization of photosensitive commercial UV filters [27] and in the controlled release of bioactive thymine photo-dimers useful for the activation of alert cellular pathway in damaged cells [28]. We report here the synthesis of novel ferulic acid and palmitic acid esters of 3-*O*-ethyl ascorbic acid by lipase B from *Candida antarctica* immobilized on LNPs and their successive encapsulation inside LNPs prepared from kraft lignin by solvent shift procedure. 3-*O*-ethyl ascorbic acid is a stabilized derivative of **AA** widely applied in cosmetic formulations [29], due to the selective protection of the most acidic C3 hydroxyl group (pK_a_ = 4.25) in the molecule [30]. In addition, ferulic acid and palmitic acid were selected since their esters are well recognized to have high stability [31,32]. LNPs showed synergistic effect with encapsulated **AA** esters improving the overall antioxidant activity and stabilizing the encapsulated compounds towards pH and temperature variations. The kinetic release of **AA** esters from LNPs showed a Fickian-like diffusion process characterized by release of the encapsulated compounds during the first 24 h. The capability of LNPs to stabilize **AA** esters, associated to the synergy effect in the antioxidant activity, highly suggested LNPs as an effective carrier of AA derivatives in cosmetic and cosmeceutical formulations.

## 2. Results and Discussion

### 2.1. Synthesis of 3-O-ethyl Ascorbic Acid Esters

Ascorbic acid esters have been previously synthesized by hazardous procedures for environment and human health, including Fischer esterification [33,34] and toxic acyl chlorides [35]. Alternative carbodiimide based coupling reactions, such as the use of N,N′-dicycloexhylcarbodiimide (DCC) and 1-ethyl-3-(3-dimethylaminopropyl)carbodiimide (EDC), were not completely deprived of environmental drawbacks [36]. Biocatalytic procedures involving lipase B from *Candida antarctica* furnished a greener procedure with both aromatic and aliphatic carboxylic acids even in organic solvent [13]. In this latter case, the esterification proceeded selectively at the primary alcoholic group of **AA** [37]. The application of lipase B from *Candida antarctica* in heterogeneous condition for the synthesis of **AA** esters is limited to the use of synthetic polyacrylic resin as a support [38]. Recently, we reported the immobilization of lipase M from *Mucor javanicus* on the surface of LNPs coated with chitosan and Concanavalin A, and its use for the synthesis of hydroxytyrosol esters in a cascade reaction with tyrosinase [39]. Here, we extended the application of this procedure (deprived of tyrosinase) for the immobilization of lipase B from *Candida antarctica* and the successive synthesis of 3-*O*-ethyl ascorbic acid esters. Briefly, the preparation of biocatalyst **I** required the following steps: (a) formation of LNPs from KL by nanoprecipitation in THF as primary solvent and water as antisolvent; (b) layer-by-layer coating of LNPs with chitosan; (c) concanavalin A deposition; and (d) immobilization of lipase. The schematic representation of the preparation of biocatalyst **I** is reported in Figure 1. The immobilization and activity of biocatalyst **I** resulted in 87% and 60%, respectively.

The kinetic parameters of biocatalyst **I** were determined by using the standard *para*-nitrophenyl laurate (*para*-NPL) assay and were compared with native lipase as a reference [40]. As reported in Table 1, biocatalyst **I** showed K_m_ value lower and V_max_ value higher than native lipase (Table 1) confirming the beneficial effect played by the immobilization process.

FE-SEM image of biocatalyst **I** showed the presence of roughly aggregated on the surface on the particles in accordance with the deposition of ConA and lipase (reported in Appendix A).

The synthesis of 3-*O*-ethyl ascorbic acid esters **4** and **5** was then performed starting from 3-*O*-ethyl ascorbic acid **1** (103 mg, 0.5 mmol) and equimolar amount of ferulic acid **2** (97.1 mg), or in alternative, palmitic acid **3** (128.2 mg), previously dissolved in 2-MeTHF (5.0 mL). 2-MeTHF was selected as reaction solvent due to the previously reported application in enzymatic transformations [41,42]. Biocatalyst **I** (5 U) was added to the solution and the mixture was gently stirred at 50 °C for 48 h under argon atmosphere in the presence of anhydrous molecular sieves (Figure 2). 3-*O*-ethyl-L-ascorbyl-6-ferulate **4** and 3-*O*-ethyl-L-ascorbyl-6-palmitate **5** were obtained in 38.8% and 47.1% yields, respectively, beside the unreacted substrate. These yields are of the same order of magnitude as those obtained by chemical procedures, such as the Mitsunobu reaction described in Appendix A.

NMR data were in accordance with the proposed structures. In particular, the ^1^H NMR spectrum of ester **4** and **5** showed the typical multiplet shift of CH_2_ of ester bond at δ ppm 4.38 and 4.29, respectively, associated to the presence of the corresponding ^13^C NMR signal at δ ppm 64.40 as reported in supporting information (Appendix A) The UV-vis absorbance analysis of dissolved esters **4** and **5** in comparison with **1** and **2** are reported in Appendix A.

### 2.2. Encapsulation of AA Derivatives and Morphological Characterization

The schematic representation for the encapsulation protocol is reported in Figure 2. Briefly, commercially available KL and esters **4** or **5** were dissolved in pure DMI [21] (primary solvent) in order to avoid the possible degradation processes due to the presence of water [43]. The solution was successively added with milliQ water (anti-solvent) at room temperature under gentle magnetic stirring with instantaneous precipitation of LNPs-**4** and LNPs-**5** which were fast recovered by centrifugation. Note that DMI was selected as primary solvent due to its certified eco-sustainability [44]. Ascorbic acid and compounds **1** and **6** were also encapsulated to obtain LNPs-**AA**, LNPs-**1**, and LNPs-**6** as references. The encapsulation protocol was studied in different compound to LNPs ratios (ratio: 1:1, 1:5, 1:10) in order to evaluate the optimal experimental conditions.

The optimal condition corresponding to well dispersed LNPs was obtained in the 1:10 ratio, while undesired aggregates were observed under the resting experimental conditions. The loading efficiency (LE) was evaluated by HPLC analysis (original chromatograms are available in Appendix A) of the residual amounts of esters in the supernatant (see Table 2). LNPs-**4** and LNPs-**5** showed LE values higher than LNPs-**AA** and LNPs-**1** as references. The high efficacy observed in the encapsulation of esters **4** and **5** was probably due to their high lipophilicity and steric hindrance which favored the interaction with the hydrophobic core of LNPs [45].

In addition, π–π interactions between the aromatic moiety of the molecule and lignin subunits were probably operative in the case of ester **4**. Note that LNPs-**4** and LNPs-**5** also showed LE values higher LNPs-**6**, suggesting the beneficial role of the ethereal protection of the **AA** scaffold in the encapsulation procedure.

The size characterization of loaded LNPs and the empty counterpart was conducted by DLS analysis (Table 3). As a general trend, the average diameter of LNPs proportionally increased after the encapsulation by increasing the LE value. The PDI and Z-potential values remained similar to the empty LNPs.

### 2.3. Release Profile and Kinetic Release Data of Esters ***4*** and ***5*** from LNPs

The release profiles of esters **4** and **5** were studied by analyzing the UV-absorption pattern of dialyzed solutions from LNPs-**4** and LNPs-**5** suspended in Na-acetate buffer (pH values 5.4 or 3.0) at room temperature for 24 h. The pH values were selected in order to model the microenvironment of the skin [46] and of typical anti-age serum formulations [47], respectively (Figure 3A). As a general trend, the release of esters **4** and **5** was faster during the first 5 h, followed by a plateau. In addition, the release profile was favored under the most acidic conditions. Ester **4** showed slight delay with respect to ester **5**, probably due to the presence of extra π–π interactions. After 24 h, esters **4** and **5** were released in 68% and 84% at pH 3.0 and in 62% and 78% at pH 5.4, respectively.

Next, kinetic profiles of LNPs-**4** and LNPs-**5** were evaluated by fitting the release profiles data with zero-order, first-order, Higuchi, and Korsmeyerñ–Peppas kinetic models at pH 3.0 as a representative pH value (Section 3). The data plotting was expressed by using R^2^ coefficient and linear regression (Figure 3B). The Higuchi model provided the best fit (R^2^ value 0.9782 for LNPs-**4** and 0.9716 for LNPs-**5**), confirming the presence of Fickian-like diffusion-controlled process [48]. This phenomenon is specific for host compounds which are characterized by a size significantly smaller than that of the carrier matrix and where swelling and matrix dissolution are negligible. Furthermore, first 60% ester release profile were fitted in Korsmeyer-Peppas model obtaining good fit (R^2^ value 0.953 for LNPs-**4** and 0.9447 for LNPs-**5**) and “n” index lower of 0.5, suggesting a quasi-Fickian diffusion process [49].

### 2.4. Antioxidant Activity

The antioxidant activity of esters **4** and **5** was evaluated by spectrophotometric assay using 2,2-diphenyl-1-picrylhydrazyl (DPPH) as a probe. Ascorbic acid and compounds **1**–**2** and **6** were studied as references. The analysis was performed at different concentrations of the sample (500, 250, 100, 50, and 10 μg/mL) in milliQ water by monitoring the reduced form of DPPH at 517 nm.

The results were expressed in terms of IC_50_ (the amount of antioxidant compound required to inhibit the starting concentration of the DPPH radical by 50%) and antioxidant capacity index ACI (the inverse value of IC_50_). Esters **4** exhibited lower IC_50_ and higher ACI values with respect to **AA** derivatives references (Table 4, entry 2–4), confirming the contribution of ferulic acid in the antioxidant activity, while ester **5** showed a similar trend to commercial analogue **6** (Table 4, entry 5–6).

Next, the IC_50_ and ACI of LNPs-**4** and LNPs-**5** were evaluated by using empty LNPs as a reference (Table 4, entry 8–10). LNPs-**4** and LNPs-**5** showed IC_50_ values (7.1 and 12.1, respectively) lower than esters **4** and **5** alone (11.5 and 17.1, respectively). The improved antioxidant activity of esters **4** and **5** after encapsulation was probably due to the co-antioxidant effect exerted by LNPs (which showed the IC_50_ value of 19.9) in the scavenging of free radicals [50]. In order to evaluate the occurrence of a synergistic effect between lignin nanoparticles and loaded esters we applied Equation (1):(1)$${IC}_{50}ratio=\frac{{LNPs}_{{IC}_{50}}}{{eLNPs}_{{IC}_{50}}+E_{{IC}_{50}}}$$
where ${LNPs}_{{IC}_{50}}$ is the IC_50_ value of LNPs-**4** or LNPs-**5**, ${eLNPs}_{{IC}_{50}}$ is the IC_50_ value of empty LNPs and $E_{{IC}_{50}}$ is the IC_50_ value of free esters **4** and **5**. The synergistic effect is represented by a ratio value < 0.5, while the simple additive effect corresponds to ratio > 0.5. LNPs-**4** and LNPs-**5** showed a synergistic effect occurring between lignin and loaded esters (ratio value 0.23 and 0.33, respectively) probably as a consequence of the regeneration of exhausted esters by electroactive LNPs trough hydrogen atom transfer (HAT) [51]. Examples of synergistic effect between amorphous lignin and antioxidant compounds are documented [52,53], and the role of the aromatic hydroxyl groups in the HAT mechanism discussed in details in the case of LNPs [54,55].

### 2.5. Thermal Stability

The chemical stability of esters **4** and **5** and corresponding LNPs-**4** and LNPs-**5** was evaluated in sodium acetate solution (pH 3.0) and DMI as co-solvent by heating the sample at 60 °C in a standard oven for 24 h in the presence of air. Sodium acetate solution was selected since pH 3 was characteristic of cosmetic formulation containing AA [47]. Ascorbic acid, compounds **1**, **6** and empty LNPs were studied as references. After the incubation time, the sample was analyzed in order to evaluate the residual amount of the original compound by HPLC analysis. In the case of LNPs-**4** and LNPs-**5** the analysis required the disaggregation of LNPs by treatment with THF at room temperature under gentle magnetic stirring followed by filtration of the organic phase on 0.23 μm disc filter.

As expected, the almost total degradation of **AA** was observed after 24 h of treatment (Figure 4), while the AA derivatives **1** and **6** showed a moderate pH and thermal resistance (42.1% and 39.5%, respectively) [56]. Esters **4**–**6** showed an improved stability as a consequence of the protection of the sensitive primary alcohol moiety in the **AA** scaffold. In this latter case, the contemporary protection of the ene-diol moiety further increased stability. Noteworthy, LNPs-**4** and LNPs-**5** fully retained stability after the thermal/pH treatment confirming the beneficial role exerted by the encapsulation process.

## 3. Materials and Methods

### 3.1. Materials

Kraft lignin (KL) was purchased from Sigma Aldrich (St. Louis, MO, USA) and used after purification by standard procedures including alkali–acid treatment and continuous washing with deionized water. Dimethyl isosorbide (DMI), ethanol, 2,2-diphenyl-1-picrylhydrazyl (DPPH), L-ascorbic acid AA, 3-*O*-ethylascorbate **1**, ferulic acid **2**, palmitic acid **3**, 6-*O*-Palmitoyl-L-ascorbic acid **6**, diisopropyl azodicarboxylate (DIAD), triphenylphosphine (TPP), butylhydroxytoluene (BHT), *para*-4-nitrophenyl laurate (*para*-NPL), chitosan (non-animal derived, average Mw 50 kDa, CH), concanavalin A (Con A, from *Canavalia ensiformis*, Jack bean type VI), and Lipase B from *Candida antarctica* were purchased from Sigma Aldrich and used without further purification. Analytical grade tetrahydrofuran THF and 2-methyl tetrahydrofuran (2-MeTHF) were refluxed over Na–benzophenone under nitrogen, followed by distillation and storage over 4 Å molecular sieves. Deuterated and HPLC purity-grade solvents were purchased from VWR. 1H and 13C NMR spectra were obtained in MeOD_3_ using a Bruker 400 MHZ spectrometer (Bruker, Billerica, MA, USA).

### 3.2. Preparation and Kinetic Characterization of Biocatalyst **I**

Biocatalyst **I** was prepared as reported in reference [39]. Briefly, LNPs (5 mg, 1.0 mg mL^−1^) obtained by nanoprecipitation from KL in Milli-Q water were added to CH (10 mg, 1.0 mg mL^−1^ in diluted acetic acid 1.0% *v*/*v*) under slow magnetic stirring (200 rpm) for 2 h at room temperature. The LNPs/CH intermediate (1.0 mg mL^−1^) was coated with ConA (0.4 mg mL^−1^) in sodium phosphate buffer (PBS, 1.0 mL; pH 7.3, 0.1 M) at room temperature to yield LNPs/CH/ConA. This latter intermediate (1.5 mg mL^−1^) was dispersed in PBS (1.0 mL; 0.1 M, pH 7) and added of lipase B from *Candida antarctica* (25 U; 3.0 mg mL^−1^) dissolved in PBS (0.1 M; pH 7) under gentle stirring (200 rpm) at 4 °C for 24 h and the suspension was centrifuged and washed two times with PBS (pH 7, 0.1 M) to afford biocatalyst **I** (overall yield 85% in weight with respect to LNPs/CH/ConA intermediate).

The enzymatic activity and kinetic parameters of biocatalyst **I** were determined by the *para*-NPL assay [57]. In particular, the activity of lipase was evaluated by measuring the increase in absorbance (λ_max_ 405 nm) produced during the release of *para*-nitrophenol in the hydrolysis of *para*-NPL in Tris-HCl (0.1 M, pH 7.4) at 25 °C for 15 min. In a typical experiment, biocatalyst **I** (1.5 mg mL^−1^ in PBS, 0.1 M, pH 7) was added to *para*-NPL solution (0.1 mL; 20 mM; CNCH_3_: water 1:1) in Tris-HCl. The international unit of lipase activity was defined as the amount of the enzyme necessary to hydrolyze 1.0 μmol of *para*-NPL in 1.0 min at 25 °C and optimal pH. Kinetic parameters (K_m_ and V_max_) of biocatalyst **I** were determined by measuring the activity of lipase (25 U) using different concentration of *para*-NPL (from 0 to 10 mM) in Tris-HCl (pH 7.4). The kinetic parameters were analyzed by Lineweaver–Burk plots and calculated in triplicate with the GraphPad Prism 9 software.

### 3.3. Synthesis and Characterization of Esters ***4*** and ***5***

3-*O*-ethyl-L-ascorbyl-6-ferulate **4** and 3-*O*-ethyl-L-ascorbyl-6-palmitate **5** were synthesized from 3-*O*-ethyl ascorbic acid **1** and the appropriate carboxylic acid in 2-MeTHF in the presence of biocatalyst **I**. Compound **1** (103 mg, 0.5 mmol) and the appropriate carboxylic acid (1.0 eq.) were dissolved in 2-MeTHF (5.0 mL) in the presence of biocatalyst **I** (5 U), and the mixture was gentle stirred at 50 °C for 48 h under argon atmosphere in the presence of anhydrous molecular sieves. At the end of the reaction, biocatalyst **I** was recovered by centrifugation (6000 rpm), and the organic phase was evaporated under reduced pressure. The crude was dissolved in EtOAc (5.0 mL), washed with NaCl ss, dried over anhydrous Na_2_SO_4_ and evaporated under high vacuum. Esters **4** and **5** were obtained after chromatographic purification by using CH_2_Cl_2_:MeOH 95:5 as eluent.

3-*O-ethyl-L-ascorbyl-6-ferulate*
**4**. Yield: 38.8% mp: 175–181 °C, yellow solid. Elemental analysis for C_18_H_20_O_9_, expected value: C, 56.84; H, 5.30; O, 37.87; measured value: C, 56.80; H, 5.30; O, 37.83. ^1^H NMR (MeOD_3_, 400 MHz): δ_H_ 7.63 (1H, d, J = 16Hz, H11), 7.19 (1H, s, H10), 7.08 (1H, d, J = 6.4 Hz, H9), 6.83 (1H, d, J = 16Hz, H8), 6.34 (1H, d, J = 15.88 Hz, H7), 4.38 (1H, d, J = 3.28 Hz, H6), 4.36–4.31 (4H, m, H5 + H4), 3.90 (3H, s, H3), 3.67 (1H, d, J = 6.96 Hz, H2) 1.40 (3H, t, H1). ^13^C NMR (MeOD_3_, 100 MHz): δ_C_ 169.58 (C18), 162.96 (C17), 150.67 (C16), 149.09 (C15), 147.96 (C14), 145.53 (C13), 126.38 (C12), 122.59 (C11), 119.06 (C10), 115.06 (C9), 114.49 (C8), 110.27 (C7), 75.33 (C6), 69.17 (C5), 67.14 (C4), 64.40 (C3), 55.04 (23), 14.40 (C1).

3-*O-ethyl-L-ascorbyl-6-palmitate*
**5**. Yield: and 47.1% mp: 121–129 °C, bright yellow solid. Elemental analysis for C_24_H_42_O_7_, expected value: C, 65.13; H, 9.57; O, 25.30; measured value: 65.10; H, 9.51; O, 25.28. ^1^H NMR (MeOD_3_, 400 MHz): δ_H_ 4.75 (1H, s, H9), 4.38–4.36 (2H, m, H8), 4.29–4.18 (2H, m, H7), 4.12–4.08 (1H, m, H6), 2.40 (2H, t, H5), 1.66–1.61 (2H, t, H4), 1.40–1.37 (3H, t, H3), 1.30 (24H, m, H2), 0.93 (3H, t, H1); ^13^C NMR (MeOD_3_, 100 MHz): δ_C_ (ppm) 173.74 (C17), 171.74 (C16), 152.59 (C15), 118.69 (C14), 75.79 (C13), 66.66 (C12), 64.40 (C11), 62.09 (C10), 33.48 (C9), 31.69 (C8), 29.39–28.81 (C7 + C6 + C5), 24.59 (C4), 22.35 (C3), 14.4 (C2), 13.07 (C1).

### 3.4. Encapsulation of AA and AA Derivatives into LNPs

The encapsulation protocol was applied for **AA**, **1**, **4**, **5**, and **6** by the solvent shifting protocol [21]. Briefly, lignin solution were prepared adding KL (20.0 mg) in DMI (2.0 mL) at 45 °C under magnetic stirring for 24 h. Successively, the appropriate compound (0.03 mmol) was added at room temperature under magnetic stirring for 6 h. MilliQ water (6.0 mL) was added to previously solution to produce the corresponding LNPs-**AA**, LNPs-**1**, LNPs-**4**, LNPs-**5**, and LNPs-**6**, which were recovered by centrifugation (2000 rpm; 20 min) and freeze dry (LABCONCO, Kansas City, MO, USA, freeze time 48 h).

### 3.5. Determination of Loading Efficiency and Kinetic of Release of LNPs-***4*** and LNPs-***5***

The loading efficiency (LE) was evaluated by HPLC analysis of the supernatant recovered from the centrifugation step by applying the Equation (2):(2)$$LE\left(\%\right)=\frac{mg\;starting\;compound-mg\;unloaded\;compound}{weight\;of\;starting\;compound}*100$$
where LE represents the amount of encapsulated compound with respect to the initial concentration of esters **4** and **5**.

The HPLC analysis was performed by Ultimate 3000 Rapid Resolution system (ThermoFisher scientific, Waltham, MA, USA) equipped with Alltima C18 (250 mm × 4.6 mm, 5 μm) column, injection volume 10 μL, mobile phase flow rate 1.0 mL/min and a detector set up at 254 nm for ascorbic acid and compounds **1**, **5**, and **6**, and 320 nm for ester **4**. The analysis was repeated in triplicate. The compounds were separated by isocratic method using 95.0% of ultra-pure formic acid and 5.0% MeOH for 25 min and quantified by using the calibration curve method.

The profile release of LNPs-**4** and LNPs-**5** was evaluated suspending the sample (10.0 mg) in Na-acetate buffer (3.0 mL, pH 3.0 and 5.4) in dialysis bag (cellulose membrane, MWCO = 1.000) inserted in the acceptor compartment (25.0 mL) under slight stirring at room temperature for 24 h. At regular intervals time, aliquots (1.50 mL) were collected from the acceptor compartment with milliQ water replacing to maintain a constant volume. The aliquots were quantified by the HPLC method previously described using the calibration curve procedure. The measurements were performed in triplicate. Data were elaborated by applying the kinetic models reported in Equations (3)–(6) [58], including zero-order (3), first-order (4), Higuchi (5), and Korsmeyer–Peppas (6) models:(3)$$Q_{0}-Q_{t}=K_{0t}$$
(4)$$\log C=\log C_{0}-\frac{K_{t}}{2.303}$$
(5)$$\frac{M_{t}}{M_{\infty }}=K_{H}*t^{\frac{1}{2}}$$
(6)$$\frac{M_{t}}{M_{\infty }}=K_{KP}*t^{nKP}$$
where Q_0_: the starting amount of compound dissolved in the solution; Q_t_: amount of released compound at time (t); K_0_: constant of zero-order release express in units of concentration/time; K: constant of first-order release presented in units of time^−1^; C_0_: starting concentration of compound; K_H_: constant of the Higuchi model; M_t_: amount of released compound; M_∞_: amount of compound in dosage form; M_t_/M_∞_: amount of compound released at time t ratio of released compound ratio; KP: release constant related to the Korsmeyer–Peppas model. In the Korsmeyer–Peppas “*n*” was found by using the portion of the release curve where M_t_/M_∞_ < 0.6.

### 3.6. Average Size and ζ-Potential Determination

The hydrodynamic diameter and ζ-potential values were evaluated in H_2_O by dynamic light scattering (DLS) using a Zetasizer Nano ZS (Malvern Instruments, Malvern, UK) apparatus, equipped with He-Ne laser (633 nm, fixed scattering angle of 173°, 25 °C). Measurements were performed in triplicate at 25 °C.

### 3.7. FE-SEM Analyses

FE-SEM was carried out by ZEISS Gemini 500 (Oberkochen, Germany) at 5 kV. The appropriate compound in water (20 μL) was deposited on a coverslip coated with gold (AGAR Auto Sputter Coated) and dehydrated in air. The coverslip was mounted on the stub with conductive carbon glue and a thin film (5.0 nm) of chromium was deposited using the Sputter QUORUM Q 150T ES plus (Laughton, United Kingdom) to make the sample sufficiently conductive for measurement purposes.

### 3.8. Antioxidant Activity

The antioxidant activity was evaluated by the 2,2′-diphenyl picrylhydrazyl (DPPH) radical scavenging assay. The test was performed starting from different amounts of sample (500, 250, 100, 50, and 10 μg/mL) in water (0.550 mL) by addition to freshly prepared DPPH solution (2.45 mL, 0.7 mM in MeOH). The spectrophotometric analysis was carried out in duplicate at 25 °C. The radical scavenging activity was monitored as a function of time by measuring the absorbance band of DPPH at 517 nm (λ_max_) until the absorbance value reached a plateau. The kinetic of the process was analyzed for each concentration tested, and the amount of DPPH remaining at the steady state was estimated. This value was used to calculate the IC_50_ (concentration of the substrate that causes 50% loss of DPPH activity). Data were expressed as percentage of inhibition of DPPH activity based on Equation (7):(7)$$\mathrm{Inhibition}\;\mathrm{of}\;\mathrm{DPPH}\;(\%):\;\frac{\left(Abs\;control-Abs\;sample\right)}{Abs\;control}\times 100$$
where “Abs control” is the absorbance of the DPPH solution in the absence of the sample. The measurements were performed in triplicate.

### 3.9. Thermal Stability Assay

The free and encapsulated AA esters (1.0 mg/mL) were incubated in Na-acetate aqueous (pH 3.0) solution in presence of DMI (7.0% *v*/*v*) as co-solvent into capped glass vials and placed in standard oven at 60 °C for 24 h. After incubation time, the residual compound was recovered by liquid-liquid extraction and quantified by HPLC analysis. In the case of LNPs-**4** and LNPs-**5** the analysis required the initial disaggregation of LNPs by treatment with THF (5.0 mL) at room temperature under gentle magnetic stirring followed by filtration of the organic phase on 0.23 μm disc filter.

## 4. Conclusions

The novel AA esters, 3-*O*-ethyl-L-ascorbyl-6-ferulate and 3-*O*-ethyl-L-ascorbyl-6-palmitate, were successfully synthesized by a sustainable procedure including lipase B from *Candida antarctica* immobilized on the surface of LNPs coated with chitosan and concanavalin A as selective spacer for the enzyme. The esters were efficiently encapsulated inside LNPs from kraft lignin by eco-sustainable solvent shifting procedure. After the encapsulation process, loaded LNPs showed improved thermal and pH stability, 3-*O*-ethyl-L-ascorbyl-6-ferulate and 3-*O*-ethyl-L-ascorbyl-6-palmitate being preserved toward degradation with respect to free **AA** and other AA derivatives as references. The best fitting of release profile of novel AA esters from LNPs with Higuchi model suggest a Fickian-diffusive mechanism reaching up to approximately 70% of release of the desired compound after 24 h. Noteworthy, the role of LNPs was not limited to improve the stability of AA esters, since the overall antioxidant activity of loaded LNPs was higher than the sum of that of the single components, highlighting the occurrence of a synergistic effect. This unexpected result was probably due to the electrochemical properties of LNPs deriving from HOMO-LUMO electron transfer process between the aromatic sub-units. A similar effect was previously observed in the case of LNPs containing chito-oligosaccharides [59]. Overall, the encapsulation of AA esters inside LNPs appears to be a new effective entry for the stabilization and activation of AA in cosmetic and cosmeceutical formulations subjected to temperature and pH stress.

## Data Availability

No new data were created or analyzed in this study. Data sharing is not applicable to this article.

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
