# Peer review of "Biosynthesis of Novel Ascorbic Acid Esters and Their Encapsulation in Lignin Nanoparticles as Carriers and Stabilizing Systems"

_ijms, 2023, doi:10.3390/ijms24109044_

Round 1

Reviewer 1 Report

Dear auhors,

This study reported an approach to encapsulating vitamin C derivatives into lignin particles, which was successfully prepared. The results of this research include new discoveries related to cosmetics and other healthcare applications. However, I think that many of the following revisions including are necessary for this article to meet the journal's criteria.

Comment 1

Throughout the paper, the authors use the term "nanoparticles" to describe the particles produced in this study. However, the particles prepared in this study are "submicron particles" because their size is 200 to 300 nm.

Endocytosis: The Nanoparticle and Submicron Nanocompounds Gateway into the Cell

https://www.mdpi.com/1999-4923/12/4/371

Nanoparticles (NPs) and submicron particles (named both from this point as NPs to abbreviate) are defined as materials with nanometric sizes (1–100 and 100–1000 nm, respectively) that interact with biological systems in an unusual way because of their high surface to volume ratio.”

Therefore, the particles prepared in this study should not be called "nanoparticles" but should be described as "submicron particles.

Comment 2

Throughout the paper, abbreviations appearing in each figure or table should be explained in the footnotes of each figure or table, even if they are already explained in the text.

Comment 3

L 162-163, for results of %LE, data should be expressed in MEAN±SD. With n=1 results, the reader may think that the difference between the each sample is merely experimental variability.

Comment 4

L 173-175, mean particle size and zeta potential data should be presented as Mean ± SD, not Mean. Data in many reports of nanoparticle and submicron particle preparations are described as Mean ± SD.

Comment 5

L 242-243, for results of IC50 of DPPH, at least these evaluations should be expressed in MEAN±SD with n=4 or more. With n=1 results, the reader may think that the difference between the each sample is merely experimental variability.

Comment 6

L 305, authors noted “calculated in triplicate with the GraphPad software”. Information on the company and product name of the “GraphPad software” used in this study is required. Is this software GraphPad Prism9 ?

Comment 7

L 333-340, AA is unstable in water, and some reports indicate that most of it decomposed in water in one day.

Ascorbic Acid Derivatives as Potential Substitutes for Ascorbic Acid To Reduce Color Degradation of Drinks Containing Ascorbic Acid and Anthocyanins from Natural Extract

https://pubs.acs.org/doi/10.1021/acs.jafc.9b05049

Under the present encapsulation conditions, is there a risk that more than half of the sample was decomposed during the 6 hours under magnetic stirring at room temperature?

Data of stability over time for each of AA, 1, 4, 5, and 6 in water should be presented.

Comment 8

L 342-381, It is essential to add new chromatograms of the HPLC analysis as figures. It is also necessary to explain which peaks in the chromatogram correspond to the target compound. Furthermore, the authors chose detection wavelengths of 254 nm, 280 nm, and 320 nm. What wavelengths of detection were used for the chromatograms of esters 4 and 5, respectively? No information is available.

Comment 9

L 397-412, many reports have indicated that a single DPPH assay alone is not sufficient in evaluating antioxidative ability.

Antioxidant-Activity-Guided Purification and Separation of Octocrylene from Saussurea heteromalla

https://bmcresnotes.biomedcentral.com/articles/10.1186/s13104-015-1618-6

Vitamin E: Regulatory Redox Interactions

https://iubmb.onlinelibrary.wiley.com/doi/full/10.1002/iub.2008

At least three different antioxidant assays should be need to demonstrate the authors' experimental goals regarding antioxidant effects.

Comment 10

L 414-421, in this study, authors incubated with sodium acetate solution (pH 3.0). What was the reason for setting only this pH? If future applications for cosmetics and other biological products are to be considered, they should be examined under a variety of pH conditions.

For example, the pH of healthy skin is said to be about 5.

The pH of the Skin Surface and Its Impact on the Barrier Function

https://karger.com/spp/article/19/6/296/295519/The-pH-of-the-Skin-Surface-and-Its-Impact-on-the

Comment 11

Regarding supplemental materials, the NMR chart in the supplemental material is more complex because of the chart alone. For example, the chemical structure should be included in the chart, as in the ACS standard, and which sites correspond to which peaks should be presented in the chart.

NMR Guidelines for ACS Journals

Please check the English text again.

Author Response

Response to Reviewer 1 Comments

Q1: Throughout the paper, the authors use the term "nanoparticles" to describe the particles produced in this study. However, the particles prepared in this study are "submicron particles" because their size is 200 to 300 nm.

Endocytosis: The Nanoparticle and Submicron Nanocompounds Gateway into the Cell

https://www.mdpi.com/1999-4923/12/4/371

“Nanoparticles (NPs) and submicron particles (named both from this point as NPs to abbreviate) are defined as materials with nanometric sizes (1–100 and 100–1000 nm, respectively) that interact with biological systems in an unusual way because of their high surface to volume ratio.”

Therefore, the particles prepared in this study should not be called "nanoparticles" but should be described as "submicron particles.”

A1: The classification of nanoparticles on the basis of their diameter value is a matter of debate. We agree with the fact that in the specific manuscript suggested by the reviewer (https://www.mdpi.com/1999-4923/12/4/371) particles with an average diameter of 200-300 nm are defined as “submicron particles” instead of “nanoparticles”. On the other hand, a very large number of other manuscripts describe the particles with an average diameter in the range comprised between 200 and 300 nm as “nanoparticles” (see just for some representative examples:        

1)https://doi.org/10.1016/j.ijbiomac.2022.07.046; 2)https://doi.org/10.1016/j.indcrop.2021.114012; 3)https://doi.org/10.3390/app10144910

For this reason, we prefer (if possible) to retain the original term “nanoparticles” in the manuscript.    

Q2: Throughout the paper, abbreviations appearing in each figure or table should be explained in the footnotes of each figure or table, even if they are already explained in the text.

A2: Thanks, the abbreviations are now explained in footnotes of tables and figures as suggested.

Q3: L 162-163, for results of %LE, data should be expressed in MEAN±SD. With n=1 results, the reader may think that the difference between the each sample is merely experimental variability.

A3: We agree with the reviewer comment. Experimental LE Data are now reported including the respective SD.

Q4: L 173-175, mean particle size and zeta potential data should be presented as Mean ± SD, not Mean. Data in many reports of nanoparticle and submicron particle preparations are described as Mean ± SD.

A4: We agree with the reviewer. The particle size and zeta potential data have been improved accordingly.

Q5: L 242-243, for results of IC50 of DPPH, at least these evaluations should be expressed in MEAN±SD with n=4 or more. With n=1 results, the reader may think that the difference between the each sample is merely experimental variability.

A5: We agree with the reviewer. The data of IC50 have been improved by adding the MEAN±SD, accordingly.

Q6: L 305, authors noted “calculated in triplicate with the GraphPad software”. Information on the company and product name of the “GraphPad software” used in this study is required. Is this software GraphPad Prism9?

A6: The software was GraphPad Prism 9. The product name was modified accordingly.

Q7: L 333-340, AA is unstable in water, and some reports indicate that most of it decomposed in water in one day.

Ascorbic Acid Derivatives as Potential Substitutes for Ascorbic Acid To Reduce Color Degradation of Drinks Containing Ascorbic Acid and Anthocyanins from Natural Extract

https://pubs.acs.org/doi/10.1021/acs.jafc.9b05049

Under the present encapsulation conditions, is there a risk that more than half of the sample was decomposed during the 6 hours under magnetic stirring at room temperature?

Data of stability over time for each of AA, 1, 4, 5, and 6 in water should be presented.

A7: It is important to note, that in the reported procedure (experimental part), AA and compounds 1, 4, 5, and 6 are dissolved only in pure dimethyilisosorbide (in the absence of water) and lignin for 6h. Thus, in this step, water was not available for degradative processes.  The presence of water in the system was limited to the few seconds required for the precipitation of nanoparticles during the next step, that is the addition of the antisolvent to dimethyilisosorbide, followed by removal of the overall liquid phase by centrifugation. The text (paragraph 2.2) has been modified accordingly in order to better clarify the details of the procedure (the reference suggested by the reviewer was also introduced): “Briefly, commercially available KL and esters 4 or 5 were dissolved in pure DMI (primary solvent) in order to avoid the possible degradation processes due to the presence of water. The solution was successively added with milliQ water (anti-solvent) at room temperature under gentle magnetic stirring with instantaneous precipitation of LNPs-4 and LNPs-5 which were fast recovered by centrifugation.” In addition, the thermal stability of AA and compounds 1, 4, 5, and 6 in water is reported in paragraph 2.5.

Q8: L 342-381, It is essential to add new chromatograms of the HPLC analysis as figures. It is also necessary to explain which peaks in the chromatogram correspond to the target compound. Furthermore, the authors chose detection wavelengths of 254 nm, 280 nm, and 320 nm. What wavelengths of detection were used for the chromatograms of esters 4 and 5, respectively? No information is available.

A8: Original HPLC chromatograms of esters 4 and 5 have been introduced in the Supporting information (as Figure S6) as requested. The following sentence has been introduced in the text to furnish this indication: “original chromatograms are available in Figure S6. In addition, the wavelengths of detection used for the chromatograms of esters 4 and 5 were 320 nm and 254 nm, respectively. This information is reported in the revised Experimental section version of the manuscript as follow: “…and a detector set up at 254 nm for ascorbic acid and compounds 1, 5 and 6, and 320 nm for ester 4”.  

Q9: L 397-412, many reports have indicated that a single DPPH assay alone is not sufficient in evaluating antioxidative ability.

Antioxidant-Activity-Guided Purification and Separation of Octocrylene from Saussurea heteromalla

https://bmcresnotes.biomedcentral.com/articles/10.1186/s13104-015-1618-6

Vitamin E: Regulatory Redox Interactions

https://iubmb.onlinelibrary.wiley.com/doi/full/10.1002/iub.2008

At least three different antioxidant assays should be need to demonstrate the authors' experimental goals regarding antioxidant effects.

A9: We agree with the reviewer about the fact that more than one antioxidant test may be better for the evaluation of the absolute value of the antioxidant effect. On the other hand, the main interest of our study was not the absolute antioxidant value, but the comparison between the relative antioxidant capacity of selected compounds before and after the encapsulation process. This goal was effectively obtained. Since different studies in the literature report the DPPH test is the only analysis, and considering the short time requested for the revision of the manuscript (few days), we cannot perform new sets of experiments hoping that the present results (that is the comparative study) may be enough interesting for the reader.

Q10: L 414-421, in this study, authors incubated with sodium acetate solution (pH 3.0). What was the reason for setting only this pH? If future applications for cosmetics and other biological products are to be considered, they should be examined under a variety of pH conditions. For example, the pH of healthy skin is said to be about 5. The pH of the Skin Surface and Its Impact on the Barrier Function https://karger.com/spp/article/19/6/296/295519/The-pH-of-the-Skin-Surface-and-Its-Impact-on-the

A10: We thank the reviewer for this comment. The thermal stability was evaluated after incubation with sodium acetate solution (pH 3.0) since this pH value has been reported as characteristic of cosmetic formulations containing AA. The following sentence and relative reference were added in the manuscript (paragraph 2.5) to clarify this point: “Sodium acetate solution was selected since pH 3 was characteristic of cosmetic formulation containing AA.”

 Q11: Regarding supplemental materials, the NMR chart in the supplemental material is more complex because of the chart alone. For example, the chemical structure should be included in the chart, as in the ACS standard, and which sites correspond to which peaks should be presented in the chart.

NMR Guidelines for ACS Journals

A11: The chemical structure of esters 4 and 5 has been added in the NMR chart and the NMR signals assignment was improved in the text (paragraph 3.3) following the NMR Guidelines for ACS Journals, as suggested.

Reviewer 2 Report

Considering the objective/hypothesis of the study, the authors succeeded in demonstrating the dual functionality of lignin nanoparticles as a biocatalyst and as a stabilizing system of ascorbic acid derivatives. Sufficient fundamental results are given for this to be a full paper. Interesting but not new the proposal to use stabilized derivative of AA. The work is well-written and is of interest to the cosmeceutical and pharmaceutical industry. Overall, I recommend this article for publication with some revisions:

1.       At line 59, should be added “…properties of encapsulated drug or bioactive principle,”

2.       Figure SI1 is equal to SEM IMAGE OF LPNs_4. Please replace LPNs FE-SEM image.

3.       At line 143-144 the sentence should be correct in “LNPs-4 and LNPs-5 precipitated instantaneously in the colloidal suspension”.

4.       At line 148 please correct with “. ..to LPNs ratios”.  Why does all indicated ratios (1:1, 1:5 and 1:10) cause undesired aggregation phenomena?  Moreover, why indicate the ratio 1:5 if then it is stated that the best ratio is 1:2?

5.       Figure 6S is misleading because the slight brightness could be due to the optical effect of completely filled vials for LPNs-4 and LPNs-5; best to present an ATR characterization to confirm the difference and the contact angle measurements to certify the “highest lipophilicity” (line 159).

6.       Please add letter (A) in the figure 3 and references for LPNs-4 (left) and LPNs-5 (right).

7.       At line 181 please add references to sentence “The pH values were selected in order to model the microenvironment of the skin and of typical anti-age serum formulations, respectively (Figure 3, Panel A).

8.       The numeration of equations is misleading; it is better to start indicating with (1) for ??50????? (line 224) and change the other equations in order of presentation in the text.

9.       About thermal stability of empty and loaded LPNs, only one temperature (60°) at pH 3.0 is tested and not different temperature conditions. Please correct in caption of figure 4 (at line 264), in the abstract (line 16) and conclusions (line 441) removing “under pH variation” and “pH”, respectively.

10.   Please report the calibration curves indicated at line 362 and used for LE% determination and the release studies of LPNs-4 and LPNs-5.

11.   Please remove “optoelectronic” at line 436.

Author Response

Response to Reviewer 2 Comments

Q1: At line 59, should be added “…properties of encapsulated drug or bioactive principle,”

A1:  The mentioned sentence (and references) was not referred to encapsulated drugs but only to empty lignin nanoparticles in comparison to amorphous lignin. The sentence was improved to better clarify this point as follow: “Compared to amorphous lignin, after the nano-structuration process, LNPs showed im-proved antioxidant, UV shielding, and anti-microbial properties, as a consequence of the p-p interactions between the aromatic subunits of the polymer”.

Q2: Figure S1 is equal to SEM IMAGE OF LPNs_4. Please replace LPNs FE-SEM image.

A2: Thank you for the suggestions. The FE-SEM image of LNPs-4 is now introduced in the text.

Q3: At line 143-144 the sentence should be correct in “LNPs-and LNPs-precipitated instantaneously in the colloidal suspension”.

A3: Thanks, the sentence was modified as follow to better clarify the details of the procedure: “Briefly, commercially available KL and esters 4 or 5 were dissolved in pure DMI (primary solvent) in order to avoid the possible degradation processes due to the presence of water. The solution was successively added with milliQ water (anti-solvent) at room temperature under gentle magnetic stirring with instantaneous precipitation of LNPs-4 and LNPs-5 which were fast recovered by centrifugation”.

Q4-1: At line 148 please correct with “. ..to LPNs ratios”. 

A4-1: Thanks for the suggestions. The sentence was corrected as requested. 

Moreover, why indicate the ratio 1:5 if then it is stated that the best ratio is 1:2?

Q4-2: Why does all indicated ratios (1:1, 1:5 and 1:10) cause undesired aggregation phenomena?

A4-2 About the selected compound to LNPs ratios, these parameters were studied in order to evaluate the optimal ratio value able to avoid undesired aggregation phenomena. The overall sentence has been modified as follow to furnish this information in the correct way: The encapsulation protocol was studied in different compound to LNPs ratios (ratio: 1:1, 1:5, 1:10) in order to evaluate the optimal experimental conditions. The optimal condition corresponding to well dispersed LNPs was obtained in the 1:10 ratio, while undesired aggregates were observed under the resting experimental conditions.”

Q4-3 Moreover, why indicate the ratio 1:5 if then it is stated that the best ratio is 1:2?

A4-3 We are sorry for the misleading. As previously stated, the best ratio was 1:10, the erroneous 1:2 value has been modified.

Q5: Figure S6 is misleading because the slight brightness could be due to the optical effect of completely filled vials for LPNs-4 and LPNs-5; best to present an ATR characterization to confirm the difference and the contact angle measurements to certify the “highest lipophilicity” (line 159).

A5: The sentence referring to brightness of LPNs-4 and LPNs-5 and the corresponding Figure have been removed from the text and Supporting Information as suggested. About lipophilicity, we used this parameter as referred to original esters 4 and 5, and not to LNPs-4 and LNPs-5. The original sentence has been modified as follow to better clarify this point “The high efficacy observed in the encapsulation of esters 4 and 5 was probably due to their high lipophilicity and steric hindrance which favored the interaction with the hydrophobic core of LNPs.”

Q6: Please add letter (A) in the figure 3 and references for LPNs-4 (left) and LPNs-5 (right).

A6: Thanks. The figure was modified accordingly.

Q7: At line 181 please add references to sentence “The pH values were selected in order to model the microenvironment of the skin and of typical anti-age serum formulations, respectively (Figure 3, Panel A).

A7: The new references have been added as requested.

Q8: The numeration of equations is misleading; it is better to start indicating with (1) for ??50????? (line 224) and change the other equations in order of presentation in the text.

A8: The numeration of equations was modified as suggested.

Q9: About thermal stability of empty and loaded LPNs, only one temperature (60°) at pH 3.0 is tested and not different temperature conditions. Please correct in caption of figure 4 (at line 264), in the abstract (line 16) and conclusions (line 441) removing “under pH variation” and “pH”, respectively.

A9: Thanks. The text was modified accordingly.

Q10Please report the calibration curves indicated at line 362 and used for LE% determination and the release studies of LPNs-4 and LPNs-5.

A10: The calibration curves have been added in supporting information.

Q11: Please remove “optoelectronic” at line 436.

A11: The word was removed from the sentence as requested.

Round 2

Reviewer 1 Report

Dear authors,

Thank you for considering my comments.